# Mondrian Forests: Efficient Online Random Forests

**Balaji Lakshminarayanan**
Gatsby Unit
University College London

**Daniel M. Roy**
Department of Engineering
University of Cambridge

**Yee Whye Teh**
Department of Statistics
University of Oxford

## Abstract

Ensembles of randomized decision trees, usually referred to as *random forests*, are widely used for classification and regression tasks in machine learning and statistics. Random forests achieve competitive predictive performance and are computationally efficient to train and test, making them excellent candidates for real-world prediction tasks. The most popular random forest variants (such as Breiman's random forest and extremely randomized trees) operate on batches of training data. Online methods are now in greater demand. Existing online random forests, however, require more training data than their batch counterpart to achieve comparable predictive performance. In this work, we use Mondrian processes (Roy and Teh, 2009) to construct ensembles of random decision trees we call *Mondrian forests*. Mondrian forests can be grown in an incremental/online fashion and remarkably, the distribution of online Mondrian forests is the same as that of batch Mondrian forests. Mondrian forests achieve competitive predictive performance comparable with existing online random forests and periodically re-trained batch random forests, while being more than an order of magnitude faster, thus representing a better computation vs accuracy tradeoff.

## 1 Introduction

Despite being introduced over a decade ago, random forests remain one of the most popular machine learning tools due in part to their accuracy, scalability, and robustness in real-world classification tasks [3]. (We refer to [6] for an excellent survey of random forests.) In this paper, we introduce a novel class of random forests—called *Mondrian forests* (MF), due to the fact that the underlying tree structure of each classifier in the ensemble is a so-called *Mondrian process*. Using the properties of Mondrian processes, we present an efficient *online* algorithm that agrees with its batch counterpart at each iteration. Not only are online Mondrian forests faster and more accurate than recent proposals for online random forest methods, but they nearly match the accuracy of state-of-the-art *batch* random forest methods trained on the same dataset.

The paper is organized as follows: In Section 2, we describe our approach at a high-level, and in Sections 3, 4, and 5, we describe the tree structures, label model, and incremental updates/predictions in more detail. We discuss related work in Section 6, demonstrate the excellent empirical performance of MF in Section 7, and conclude in Section 8 with a discussion about future work.

## 2 Approach

Given $N$ labeled examples $(\boldsymbol{x}_1, y_1), \ldots, (\boldsymbol{x}_N, y_N) \in \mathbb{R}^D \times \mathcal{Y}$ as training data, our task is to predict labels $y \in \mathcal{Y}$ for unlabeled test points $\boldsymbol{x} \in \mathbb{R}^D$. We will focus on multi-class classification where $\mathcal{Y} := \{1, \ldots, K\}$, however, it is possible to extend the methodology to other supervised learning tasks such as regression. Let $\boldsymbol{X}_{1:n} := (\boldsymbol{x}_1, \ldots, \boldsymbol{x}_n)$, $Y_{1:n} := (y_1, \ldots, y_n)$, and $\mathcal{D}_{1:n} := (\boldsymbol{X}_{1:n}, Y_{1:n})$.

A Mondrian forest classifier is constructed much like a random forest: Given training data $\mathcal{D}_{1:N}$, we sample an independent collection $T_1, \ldots, T_M$ of so-called Mondrian trees, which we will describe in the next section. The prediction made by each Mondrian tree $T_m$ is a distribution $p_{T_m}(y|\boldsymbol{x}, \mathcal{D}_{1:N})$ over the class label $y$ for a test point $\boldsymbol{x}$. The prediction made by the Mondrian forest is the average $\frac{1}{M} \sum_{m=1}^M p_{T_m}(y|\boldsymbol{x}, \mathcal{D}_{1:N})$ of the individual tree predictions. As $M \to \infty$, the

average converges at the standard rate to the expectation $\mathbb{E}_{T \sim \mathrm{MT}(\lambda, \mathcal{D}_{1:N})}[p_T(y|\boldsymbol{x}, \mathcal{D}_{1:N})]$, where $\mathrm{MT}(\lambda, \mathcal{D}_{1:N})$ is the distribution of a Mondrian tree. As the limiting expectation does not depend on $M$, we would not expect to see overfitting behavior as $M$ increases. A similar observation was made by Breiman in his seminal article [2] introducing random forests. Note that the averaging procedure above is ensemble model combination and *not* Bayesian model averaging.

In the online learning setting, the training examples are presented one after another in a sequence of trials. Mondrian forests excel in this setting: at iteration $N + 1$, each Mondrian tree $T \sim \mathrm{MT}(\lambda, \mathcal{D}_{1:N})$ is updated to incorporate the next labeled example $(\boldsymbol{x}_{N+1}, y_{N+1})$ by sampling an extended tree $T'$ from a distribution $\mathrm{MTx}(\lambda, T, \mathcal{D}_{N+1})$. Using properties of the Mondrian process, we can choose a probability distribution $\mathrm{MTx}$ such that $T' = T$ on $\mathcal{D}_{1:N}$ and $T'$ is distributed according to $\mathrm{MT}(\lambda, \mathcal{D}_{1:N+1})$, i.e.,

$$\begin{aligned} T &\sim \mathrm{MT}(\lambda, \mathcal{D}_{1:N}) \\ T' \mid T, \mathcal{D}_{1:N+1} &\sim \mathrm{MTx}(\lambda, T, \mathcal{D}_{N+1}) \end{aligned} \quad implies \quad T' \sim \mathrm{MT}(\lambda, \mathcal{D}_{1:N+1}). \quad (1)$$

Therefore, the distribution of Mondrian trees trained on a dataset in an incremental fashion is the same as that of Mondrian trees trained on the same dataset in a batch fashion, irrespective of the order in which the data points are observed. To the best of our knowledge, none of the existing online random forests have this property. Moreover, we can sample from $\mathrm{MTx}(\lambda, T, \mathcal{D}_{N+1})$ efficiently: the complexity scales with the depth of the tree, which is typically logarithmic in $N$.

While treating the online setting as a sequence of larger and larger batch problems is normally computationally prohibitive, this approach can be achieved efficiently with Mondrian forests. In the following sections, we define the Mondrian tree distribution $\mathrm{MT}(\lambda, \mathcal{D}_{1:N})$, the label distribution $p_T(y|\boldsymbol{x}, \mathcal{D}_{1:N})$, and the update distribution $\mathrm{MTx}(\lambda, T, \mathcal{D}_{N+1})$.

## 3 Mondrian trees

For our purposes, a **decision tree** on $\mathbb{R}^D$ will be a hierarchical, binary partitioning of $\mathbb{R}^D$ and a rule for predicting the label of test points given training data. More carefully, a **rooted, strictly-binary tree** is a finite tree $\mathsf{T}$ such that every node in $\mathsf{T}$ is either a leaf or internal node, and every node is the child of exactly one parent node, except for a distinguished root node, represented by $\epsilon$, which has no parent. Let $\mathsf{leaves}(\mathsf{T})$ denote the set of leaf nodes in $\mathsf{T}$. For every internal node $j \in \mathsf{T} \setminus \mathsf{leaves}(\mathsf{T})$, there are exactly two children nodes, represented by $\mathsf{left}(j)$ and $\mathsf{right}(j)$. To each node $j \in \mathsf{T}$, we associate a block $B_j \subseteq \mathbb{R}^D$ of the input space as follows: We let $B_\epsilon := \mathbb{R}^D$. Each internal node $j \in \mathsf{T} \setminus \mathsf{leaves}(\mathsf{T})$ is associated with a **split** $(\delta_j, \xi_j)$, where $\delta_j \in \{1, 2, \ldots, D\}$ denotes the dimension of the split and $\xi_j$ denotes the location of the split along dimension $\delta_j$. We then define

$$B_{\mathsf{left}(j)} := \{\boldsymbol{x} \in B_j : x_{\delta_j} \leq \xi_j\} \quad \text{and} \quad B_{\mathsf{right}(j)} := \{\boldsymbol{x} \in B_j : x_{\delta_j} > \xi_j\}. \quad (2)$$

We may write $B_j = (\ell_{j1}, u_{j1}] \times \ldots \times (\ell_{jD}, u_{jD}]$, where $\ell_{jd}$ and $u_{jd}$ denote the *l*ower and *u*pper bounds, respectively, of the rectangular block $B_j$ along dimension $d$. Put $\boldsymbol{\ell}_j = \{\ell_{j1}, \ell_{j2}, \ldots, \ell_{jD}\}$ and $\mathbf{u}_j = \{u_{j1}, u_{j2}, \ldots, u_{jD}\}$. The decision tree structure is represented by the tuple $T = (\mathsf{T}, \boldsymbol{\delta}, \boldsymbol{\xi})$. We refer to Figure 1(a) for a simple illustration of a decision tree.

It will be useful to introduce some additional notation. Let $\mathsf{parent}(j)$ denote the parent of node $j$. Let $N(j)$ denote the indices of training data points at node $j$, i.e., $N(j) = \{n \in \{1, \ldots, N\} : \boldsymbol{x}_n \in B_j\}$. Let $\mathcal{D}_{N(j)} = \{\boldsymbol{X}_{N(j)}, Y_{N(j)}\}$ denote the features and labels of training data points at node $j$. Let $\ell_{jd}^x$ and $u_{jd}^x$ denote the lower and upper bounds of training data points (hence the superscript $x$) respectively in node $j$ along dimension $d$. Let $B_j^x = (\ell_{j1}^x, u_{j1}^x] \times \ldots \times (\ell_{jD}^x, u_{jD}^x] \subseteq B_j$ denote the smallest rectangle that encloses the training data points in node $j$.

### 3.1 Mondrian process distribution over decision trees

Mondrian processes, introduced by Roy and Teh [19], are families $\{\mathcal{M}_t : t \in [0, \infty)\}$ of random, hierarchical binary partitions of $\mathbb{R}^D$ such that $\mathcal{M}_t$ is a refinement of $\mathcal{M}_s$ whenever $t > s$.[1] Mondrian processes are natural candidates for the partition structure of random decision trees, but Mondrian

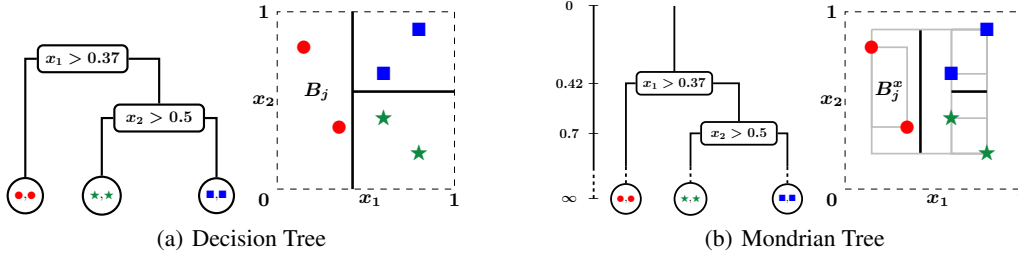

(a) Decision Tree           (b) Mondrian Tree

**Figure 1:** Example of a decision tree in $[0,1]^2$ where $x_1$ and $x_2$ denote horizontal and vertical axis respectively: Figure 1(a) shows tree structure and partition of a decision tree, while Figure 1(b) shows a Mondrian tree. Note that the Mondrian tree is embedded on a vertical time axis, with each node associated with a time of split and the splits are committed only within the range of the training data in each block (denoted by gray rectangles). Let $j$ denote the left child of the root: $B_j = (0, 0.37] \times (0, 1]$ denotes the block associated with red circles and $B_j^x \subseteq B_j$ is the smallest rectangle enclosing the two data points.

processes on $\mathbb{R}^D$ are, in general, infinite structures that we cannot represent all at once. Because we only care about the partition on a finite set of observed data, we introduce **Mondrian trees**, which are restrictions of Mondrian processes to a finite set of points. A Mondrian tree $T$ can be represented by a tuple $(\mathsf{T}, \boldsymbol{\delta}, \boldsymbol{\xi}, \boldsymbol{\tau})$, where $(\mathsf{T}, \boldsymbol{\delta}, \boldsymbol{\xi})$ is a decision tree and $\boldsymbol{\tau} = \{\tau_j\}_{j \in \mathsf{T}}$ associates a time of split $\tau_j \geq 0$ with each node $j$. Split times increase with depth, i.e., $\tau_j > \tau_{\mathsf{parent}(j)}$. We abuse notation and define $\tau_{\mathsf{parent}(\epsilon)} = 0$.

Given a non-negative *lifetime* parameter $\lambda$ and training data $\mathcal{D}_{1:n}$, the generative process for sampling Mondrian trees from $\mathrm{MT}(\lambda, \mathcal{D}_{1:n})$ is described in the following two algorithms:

---

**Algorithm 1** SampleMondrianTree$(\lambda, \mathcal{D}_{1:n})$

1: Initialize: $\mathsf{T} = \emptyset$, leaves$(\mathsf{T}) = \emptyset$, $\boldsymbol{\delta} = \emptyset$, $\boldsymbol{\xi} = \emptyset$, $\boldsymbol{\tau} = \emptyset$, $N(\epsilon) = \{1, 2, \ldots, n\}$
2: SampleMondrianBlock$(\epsilon, \mathcal{D}_{N(\epsilon)}, \lambda)$        ▷ *Algorithm 2*

---

**Algorithm 2** SampleMondrianBlock$(j, \mathcal{D}_{N(j)}, \lambda)$

1: Add $j$ to $\mathsf{T}$
2: For all $d$, set $\ell_{jd}^x = \min(\boldsymbol{X}_{N(j),d}), u_{jd}^x = \max(\boldsymbol{X}_{N(j),d})$    ▷ *dimension-wise* min *and* max
3: Sample $E$ from exponential distribution with rate $\sum_d (u_{jd}^x - \ell_{jd}^x)$
4: **if** $\tau_{\mathsf{parent}(j)} + E < \lambda$ **then**             ▷ *j is an internal node*
5:      Set $\tau_j = \tau_{\mathsf{parent}(j)} + E$
6:      Sample split dimension $\delta_j$, choosing $d$ with probability proportional to $u_{jd}^x - \ell_{jd}^x$
7:      Sample split location $\xi_j$ uniformly from interval $[\ell_{j\delta_j}^x, u_{j\delta_j}^x]$
8:      Set $N(\mathsf{left}(j)) = \{n \in N(j) : \boldsymbol{X}_{n,\delta_j} \leq \xi_j\}$ and $N(\mathsf{right}(j)) = \{n \in N(j) : \boldsymbol{X}_{n,\delta_j} > \xi_j\}$
9:      SampleMondrianBlock$(\mathsf{left}(j), \mathcal{D}_{N(\mathsf{left}(j))}, \lambda)$
10:     SampleMondrianBlock$(\mathsf{right}(j), \mathcal{D}_{N(\mathsf{right}(j))}, \lambda)$
11: **else**                                  ▷ *j is a leaf node*
12:      Set $\tau_j = \lambda$ and add $j$ to leaves$(\mathsf{T})$

---

The procedure starts with the root node $\epsilon$ and recurses down the tree. In Algorithm 2, we first compute the $\boldsymbol{\ell}_\epsilon^x$ and $\mathbf{u}_\epsilon^x$ i.e. the lower and upper bounds of $B_\epsilon^x$, the smallest rectangle enclosing $\boldsymbol{X}_{N(\epsilon)}$. We sample $E$ from an exponential distribution whose rate is the so-called linear dimension of $B_\epsilon^x$, given by $\sum_d (u_{\epsilon d}^x - \ell_{\epsilon d}^x)$. Since $\tau_{\mathsf{parent}(\epsilon)} = 0$, $E + \tau_{\mathsf{parent}(\epsilon)} = E$. If $E \geq \lambda$, the time of split is not within the lifetime $\lambda$; hence, we assign $\epsilon$ to be a leaf node and the procedure halts. (Since $\mathbb{E}[E] = 1/\left(\sum_d (u_{jd}^x - \ell_{jd}^x)\right)$, bigger rectangles are less likely to be leaf nodes.) Else, $\epsilon$ is an internal node and we sample a split $(\delta_\epsilon, \xi_\epsilon)$ from the *uniform split distribution* on $B_\epsilon^x$. More precisely, we first sample the dimension $\delta_\epsilon$, taking the value $d$ with probability proportional to $u_{\epsilon d}^x - \ell_{\epsilon d}^x$, and then sample the split location $\xi_\epsilon$ uniformly from the interval $[\ell_{\epsilon \delta_\epsilon}^x, u_{\epsilon \delta_\epsilon}^x]$. The procedure then recurses along the left and right children.

Mondrian trees differ from standard decision trees (e.g. CART, C4.5) in the following ways: (i) the splits are sampled independent of the labels $Y_{N(j)}$; (ii) every node $j$ is associated with a split

time denoted by $\tau_j$; (iii) the lifetime parameter $\lambda$ controls the total number of splits (similar to the maximum depth parameter for standard decision trees); (iv) the split represented by an internal node $j$ holds only within $B_j^x$ and not the whole of $B_j$. No commitment is made in $B_j \setminus B_j^x$. Figure 1 illustrates the difference between decision trees and Mondrian trees.

Consider the family of distributions $\mathrm{MT}(\lambda, F)$, where $F$ ranges over all possible finite sets of data points. Due to the fact that these distributions are derived from that of a Mondrian process on $\mathbb{R}^D$ restricted to a set $F$ of points, the family $\mathrm{MT}(\lambda, \cdot)$ will be *projective*. Intuitively, projectivity implies that the tree distributions possess a type of self-consistency. In words, if we sample a Mondrian tree $T$ from $\mathrm{MT}(\lambda, F)$ and then restrict the tree $T$ to a subset $F' \subseteq F$ of points, then the restricted tree $T'$ has distribution $\mathrm{MT}(\lambda, F')$. Most importantly, projectivity gives us a consistent way to extend a Mondrian tree on a data set $\mathcal{D}_{1:N}$ to a larger data set $\mathcal{D}_{1:N+1}$. We exploit this property to incrementally grow a Mondrian tree: we instantiate the Mondrian tree on the observed training data points; upon observing a new data point $\mathcal{D}_{N+1}$, we *extend* the Mondrian tree by sampling from the conditional distribution of a Mondrian tree on $\mathcal{D}_{1:N+1}$ given its restriction to $\mathcal{D}_{1:N}$, denoted by $\mathrm{MTx}(\lambda, T, \mathcal{D}_{N+1})$ in (1). Thus, a Mondrian process on $\mathbb{R}^D$ is represented only where we have observed training data.

## 4 Label distribution: model, hierarchical prior, and predictive posterior

So far, our discussion has been focused on the tree structure. In this section, we focus on the predictive label distribution, $p_T(y|\boldsymbol{x}, \mathcal{D}_{1:N})$, for a tree $T = (\mathsf{T}, \boldsymbol{\delta}, \boldsymbol{\xi}, \boldsymbol{\tau})$, dataset $\mathcal{D}_{1:N}$, and test point $\boldsymbol{x}$. Let $\mathsf{leaf}(\boldsymbol{x})$ denote the unique leaf node $j \in \mathsf{leaves}(\mathsf{T})$ such that $\boldsymbol{x} \in B_j$. Intuitively, we want the predictive label distribution at $\boldsymbol{x}$ to be a smoothed version of the empirical distribution of labels for points in $B_{\mathsf{leaf}(\boldsymbol{x})}$ and in $B_{j'}$ for nearby nodes $j'$. We achieve this smoothing via a hierarchical Bayesian approach: every node is associated with a label distribution, and a prior is chosen under which the label distribution of a node is similar to that of its parent's. The predictive $p_T(y|\boldsymbol{x}, \mathcal{D}_{1:N})$ is then obtained via marginalization.

As is common in the decision tree literature, we assume the labels within each block are independent of $\boldsymbol{X}$ given the tree structure. For every $j \in \mathsf{T}$, let $G_j$ denote the distribution of labels at node $j$, and let $\mathcal{G} = \{G_j : j \in \mathsf{T}\}$ be the set of label distributions at all the nodes in the tree. Given $T$ and $\mathcal{G}$, the predictive label distribution at $\boldsymbol{x}$ is $p(y|\boldsymbol{x}, T, \mathcal{G}) = G_{\mathsf{leaf}(\boldsymbol{x})}$, i.e., the label distribution at the node $\mathsf{leaf}(\boldsymbol{x})$. In this paper, we focus on the case of categorical labels taking values in the set $\{1, \dots, K\}$, and so we abuse notation and write $G_{j,k}$ for the probability that a point in $B_j$ is labeled $k$.

We model the collection $G_j$, for $j \in \mathsf{T}$, as a hierarchy of normalized stable processes (NSP) [24]. A NSP prior is a distribution over distributions and is a special case of the Pitman-Yor process (PYP) prior where the concentration parameter is taken to zero [17].[2] The discount parameter $d \in (0, 1)$ controls the variation around the base distribution; if $G_j \sim \mathrm{NSP}(d, H)$, then $\mathbb{E}[G_{jk}] = H_k$ and $\mathrm{Var}[G_{jk}] = (1 - d)H_k(1 - H_k)$. We use a hierarchical NSP (HNSP) prior over $G_j$ as follows:

$$G_\epsilon | H \sim \mathrm{NSP}(d_\epsilon, H), \qquad \text{and} \qquad G_j | G_{\mathsf{parent}(j)} \sim \mathrm{NSP}(d_j, G_{\mathsf{parent}(j)}). \qquad (3)$$

This hierarchical prior was first proposed by Wood et al. [24]. Here we take the base distribution $H$ to be the uniform distribution over the $K$ labels, and set $d_j = \exp\big(-\gamma(\tau_j - \tau_{\mathsf{parent}(j)})\big)$.

Given training data $\mathcal{D}_{1:N}$, the predictive distribution $p_T(y|\boldsymbol{x}, \mathcal{D}_{1:N})$ is obtained by integrating over $\mathcal{G}$, i.e., $p_T(y|\boldsymbol{x}, \mathcal{D}_{1:N}) = \mathbb{E}_{\mathcal{G} \sim p_T(\mathcal{G}|\mathcal{D}_{1:N})}[G_{\mathsf{leaf}(\boldsymbol{x}),y}] = \overline{G}_{\mathsf{leaf}(\boldsymbol{x}),y}$, where the posterior $p_T(\mathcal{G}|\mathcal{D}_{1:N}) \propto p_T(\mathcal{G}) \prod_{n=1}^N G_{\mathsf{leaf}(\boldsymbol{x}_n),y_n}$. Posterior inference in the HNSP, i.e., computation of the posterior means $\overline{G}_{\mathsf{leaf}(\boldsymbol{x})}$, is a special case of posterior inference in the hierarchical PYP (HPYP). In particular, Teh [22] considers the HPYP with multinomial likelihood (in the context of language modeling). The model considered here is a special case of [22]. Exact inference is intractable and hence we resort to approximations. In particular, we use a fast approximation known as the interpolated Kneser-Ney (IKN) smoothing [22], a popular technique for smoothing probabilities in language modeling [13]. The IKN approximation in [22] can be extended in a straightforward fashion to the online setting, and the computational complexity of adding a new training instance is linear in the depth of the tree. We refer the reader to Appendix A for further details.

## 5   Online training and prediction

In this section, we describe the family of distributions $\text{MTx}(\lambda, T, \mathcal{D}_{N+1})$, which are used to incrementally add a data point, $\mathcal{D}_{N+1}$, to a tree $T$. These updates are based on the conditional Mondrian algorithm [19], specialized to a finite set of points. In general, one or more of the following three operations may be executed while introducing a new data point: (i) introduction of a new split 'above' an existing split, (ii) extension of an existing split to the updated extent of the block and (iii) splitting an existing leaf node into two children. To the best of our knowledge, existing online decision trees use just the third operation, and the first two operations are unique to Mondrian trees. The complete pseudo-code for incrementally updating a Mondrian tree $T$ with a new data point $\mathcal{D}$ according to $\text{MTx}(\lambda, T, \mathcal{D})$ is described in the following two algorithms. Figure 2 walks through the algorithms on a toy dataset.

---

**Algorithm 3** ExtendMondrianTree$(T, \lambda, \mathcal{D})$

1: Input: Tree $T = (\mathsf{T}, \boldsymbol{\delta}, \boldsymbol{\xi}, \boldsymbol{\tau})$, new training instance $\mathcal{D} = (\boldsymbol{x}, y)$
2: ExtendMondrianBlock$(T, \lambda, \epsilon, \mathcal{D})$               ▷ *Algorithm 4*

---

**Algorithm 4** ExtendMondrianBlock$(T, \lambda, j, \mathcal{D})$

1: Set $\mathbf{e}^\ell = \max(\boldsymbol{\ell}_j^x - \boldsymbol{x}, 0)$ and $\mathbf{e}^u = \max(\boldsymbol{x} - \mathbf{u}_j^x, 0)$      ▷ $\mathbf{e}^\ell = \mathbf{e}^u = \mathbf{0}_D$ if $\boldsymbol{x} \in B_j^x$
2: Sample $E$ from exponential distribution with rate $\sum_d (e_d^\ell + e_d^u)$
3: **if** $\tau_{\mathsf{parent}(j)} + E < \tau_j$ **then**           ▷ *introduce new parent for node $j$*
4:      Sample split dimension $\delta$, choosing $d$ with probability proportional to $e_d^\ell + e_d^u$
5:      Sample split location $\xi$ uniformly from interval $[u_{j,\delta}^x, x_\delta]$ **if** $x_\delta > u_{j,\delta}^x$ **else** $[x_\delta, \ell_{j,\delta}^x]$.
6:      Insert a new node $\tilde{\jmath}$ just above node $j$ in the tree, and a new leaf $j''$, sibling to $j$, where
7:          $\delta_{\tilde{\jmath}} = \delta, \xi_{\tilde{\jmath}} = \xi, \tau_{\tilde{\jmath}} = \tau_{\mathsf{parent}(j)} + E, \boldsymbol{\ell}_{\tilde{\jmath}}^x = \min(\boldsymbol{\ell}_j^x, \boldsymbol{x}), \mathbf{u}_{\tilde{\jmath}}^x = \max(\mathbf{u}_j^x, \boldsymbol{x})$
8:          $j'' = \mathsf{left}(\tilde{\jmath})$ **iff** $x_{\delta_{\tilde{\jmath}}} \leq \xi_{\tilde{\jmath}}$
9:      SampleMondrianBlock$\big(j'', \mathcal{D}, \lambda\big)$
10: **else**
11:      Update $\boldsymbol{\ell}_j^x \leftarrow \min(\boldsymbol{\ell}_j^x, \boldsymbol{x}), \mathbf{u}_j^x \leftarrow \max(\mathbf{u}_j^x, \boldsymbol{x})$      ▷ *update extent of node $j$*
12:      **if** $j \notin \mathsf{leaves}(\mathsf{T})$ **then**      ▷ *return if $j$ is a leaf node, else recurse down the tree*
13:          **if** $x_{\delta_j} \leq \xi_j$ **then** $\mathsf{child}(j) = \mathsf{left}(j)$ **else** $\mathsf{child}(j) = \mathsf{right}(j)$
14:          ExtendMondrianBlock$(T, \lambda, \mathsf{child}(j), \mathcal{D})$      ▷ *recurse on child containing $\mathcal{D}$*

---

In practice, random forest implementations stop splitting a node when all the labels are identical and assign it to be a leaf node. To make our MF implementation comparable, we '*pause*' a Mondrian block when all the labels are identical; if a new training instance lies within $B_j$ of a paused leaf node $j$ and has the same label as the rest of the data points in $B_j$, we continue pausing the Mondrian block. We '*un-pause*' the Mondrian block when there is more than one unique label in that block. Algorithms 9 and 10 in the supplementary material discuss versions of SampleMondrianBlock and ExtendMondrianBlock for paused Mondrians.

**Prediction using Mondrian tree**    Let $\boldsymbol{x}$ denote a test data point. If $\boldsymbol{x}$ is already 'contained' in the tree $T$, i.e., if $\boldsymbol{x} \in B_j^x$ for some leaf $j \in \mathsf{leaves}(\mathsf{T})$, then the prediction is taken to be $\overline{G}_{\mathsf{leaf}(\boldsymbol{x})}$. Otherwise, we somehow need to incorporate $\boldsymbol{x}$. One choice is to extend $T$ by sampling $T'$ from $\text{MTx}(\lambda, T, \boldsymbol{x})$ as described in Algorithm 3, and set the prediction to $\overline{G}_j$, where $j \in \mathsf{leaves}(\mathsf{T}')$ is the leaf node containing $\boldsymbol{x}$. A particular extension $T'$ might lead to an overly confident prediction; hence, we average over *every* possible extension $T'$. This integration can be carried out analytically and the computational complexity is linear in the depth of the tree. We refer to Appendix B for further details.

## 6   Related work

The literature on random forests is vast and we do not attempt to cover it comprehensively; we provide a brief review here and refer to [6] and [8] for a recent review of random forests in batch and online settings respectively. Classic decision tree induction procedures choose the best split dimension and location from all candidate splits at each node by optimizing some suitable quality criterion (e.g. information gain) in a greedy manner. In a random forest, the individual trees are randomized to de-correlate their predictions. The most common strategies for injecting randomness are (i) bagging [1] and (ii) randomly subsampling the set of candidate splits within each node.

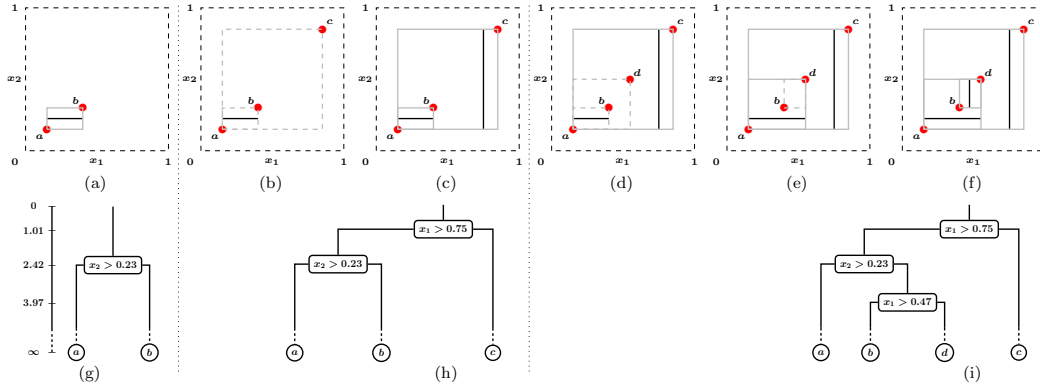

**Figure 2:** Online learning with Mondrian trees on a toy dataset: We assume that $\lambda = \infty$, $D = 2$ and add one data point at each iteration. For simplicity, we ignore class labels and denote location of training data with red circles. Figures 2(a), 2(c) and 2(f) show the partitions after the first, second and third iterations, respectively, with the intermediate figures denoting intermediate steps. Figures 2(g), 2(h) and 2(i) show the trees after the first, second and third iterations, along with a shared vertical time axis.

At iteration 1, we have two training data points, labeled as $a, b$. Figures 2(a) and 2(g) show the partition and tree structure of the Mondrian tree. Note that even though there is a split $x_2 > 0.23$ at time $t = 2.42$, we commit this split only within $B_j^x$ (shown by the gray rectangle).

At iteration 2, a new data point $c$ is added. Algorithm 3 starts with the root node and recurses down the tree. Algorithm 4 checks if the new data point lies within $B_\epsilon^x$ by computing the additional extent $\mathbf{e}^\ell$ and $\mathbf{e}^u$. In this case, $c$ does not lie within $B_\epsilon^x$. Let $R_{ab}$ and $R_{abc}$ respectively denote the small gray rectangle (enclosing $a, b$) and big gray rectangle (enclosing $a, b, c$) in Figure 2(b). While extending the Mondrian from $R_{ab}$ to $R_{abc}$, we could either introduce a new split in $R_{abc}$ outside $R_{ab}$ or extend the split in $R_{ab}$ to the new range. To choose between these two options, we sample the time of this new split: we first sample $E$ from an exponential distribution whose rate is the sum of the additional extent, i.e., $\sum_d (e_d^\ell + e_d^u)$, and set the time of the new split to $E + \tau_{\mathsf{parent}(\epsilon)}$. If $E + \tau_{\mathsf{parent}(\epsilon)} \leq \tau_\epsilon$, this new split in $R_{abc}$ can precede the old split in $R_{ab}$ and a split is sampled in $R_{abc}$ outside $R_{ab}$. In Figures 2(c) and 2(h), $E + \tau_{\mathsf{parent}(\epsilon)} = 1.01 + 0 \leq 2.42$, hence a new split $x_1 > 0.75$ is introduced. The farther a new data point $\boldsymbol{x}$ is from $B_j^x$, the higher the rate $\sum_d (e_d^\ell + e_d^u)$, and subsequently the higher the probability of a new split being introduced, since $\mathbb{E}[E] = 1/\big(\sum_d (e_d^\ell + e_d^u)\big)$. A new split in $R_{abc}$ is sampled such that it is consistent with the existing partition structure in $R_{ab}$ (i.e., the new split cannot slice through $R_{ab}$).

In the final iteration, we add data point $d$. In Figure 2(d), the data point $d$ lies within the extent of the root node, hence we traverse to the left side of the root and update $B_j^x$ of the internal node containing $\{a, b\}$ to include $d$. We could either introduce a new split or extend the split $x_2 > 0.23$. In Figure 2(e), we extend the split $x_2 > 0.23$ to the new extent, and traverse to the leaf node in Figure 2(h) containing $b$. In Figures 2(f) and 2(i), we sample $E = 1.55$ and since $\tau_{\mathsf{parent}(j)} + E = 2.42 + 1.55 = 3.97 \leq \lambda = \infty$, we introduce a new split $x_1 > 0.47$.

Two popular random forest variants in the batch setting are *Breiman-RF* [2] and *Extremely randomized trees (ERT)* [12]. Breiman-RF uses bagging and furthermore, at each node, a random $k$-dimensional subset of the original $D$ features is sampled. ERT chooses a $k$ dimensional subset of the features and then chooses one split location each for the $k$ features randomly (unlike Breiman-RF which considers all possible split locations along a dimension). ERT does not use bagging. When $k = 1$, the ERT trees are *totally randomized* and the splits are chosen independent of the labels; hence the ERT-1 method is very similar to MF in the batch setting in terms of tree induction. (Note that unlike ERT, MF uses HNSP to smooth predictive estimates and allows a test point to branch off into its own node.) Perfect random trees (PERT), proposed by Cutler and Zhao [7] for classification problems, produce totally randomized trees similar to ERT-1, although there are some slight differences [12].

Existing online random forests (ORF-Saffari [20] and ORF-Denil [8]) start with an empty tree and grow the tree incrementally. Every leaf of every tree maintains a list of $k$ candidate splits and associated quality scores. When a new data point is added, the scores of the candidate splits at the corresponding leaf node are updated. To reduce the risk of choosing a sub-optimal split based on noisy quality scores, additional hyper parameters such as the minimum number of data points at a leaf node before a decision is made and the minimum threshold for the quality criterion of the best split, are used to assess 'confidence' associated with a split. Once these criteria are satisfied at a leaf node, the best split is chosen (making this node an internal node) and its two children are the new leaf nodes (with their own candidate splits), and the process is repeated. These methods could be

memory inefficient for deep trees due to the high cost associated with maintaining candidate quality scores for the fringe of potential children [8].

There has been some work on incremental induction of decision trees, e.g. incremental CART [5], ITI [23], VFDT [11] and dynamic trees [21], but to the best of our knowledge, these are focused on learning decision trees and have not been generalized to online random forests. We do not compare MF to incremental decision trees, since random forests are known to outperform single decision trees.

Bayesian models of decision trees [4, 9] typically specify a distribution over decision trees; such distributions usually depend on $X$ and lack the projectivity property of the Mondrian process. More importantly, MF performs ensemble model combination and not Bayesian model averaging over decision trees. (See [10] for a discussion on the advantages of ensembles over single models, and [15] for a comparison of Bayesian model averaging and model combination.)

# 7    Empirical evaluation

The purpose of these experiments is to evaluate the predictive performance (test accuracy) of MF as a function of (i) fraction of training data and (ii) training time. We divide the training data into 100 mini-batches and we compare the performance of online random forests (MF, ORF-Saffari [20]) to batch random forests (Breiman-RF, ERT-$k$, ERT-1) which are trained on the same fraction of the training data. (We compare MF to dynamic trees as well; see Appendix F for more details.) Our scripts are implemented in Python. We implemented the ORF-Saffari algorithm as well as ERT in Python for timing comparisons. The scripts can be downloaded from the authors' webpages. We did not implement the ORF-Denil [8] algorithm since the predictive performance reported in [8] is very similar to that of ORF-Saffari and the computational complexity of the ORF-Denil algorithm is worse than that of ORF-Saffari. We used the Breiman-RF implementation in *scikit-learn* [16].[3]

We evaluate on four of the five datasets used in [20] — we excluded the *mushroom* dataset as even very simple logical rules achieve $> 99\%$ accuracy on this dataset.[4] We re-scaled the datasets such that each feature takes on values in the range $[0, 1]$ (by subtracting the min value along that dimension and dividing by the range along that dimension, where range $=$ max $-$ min).

As is common in the random forest literature [2], we set the number of trees $M = 100$. For Mondrian forests, we set the lifetime $\lambda = \infty$ and the HNSP discount parameter $\gamma = 10D$. For ORF-Saffari, we set num_epochs $= 20$ (number of passes through the training data) and set the other hyper parameters to the values used in [20]. For Breiman-RF and ERT, the hyper parameters are set to default values. We repeat each algorithm with five random initializations and report the mean performance. The results are shown in Figure 3. (The * in Breiman-RF* indicates *scikit-learn* implementation.)

Comparing test accuracy vs fraction of training data on *usps*, *satimages* and *letter* datasets, we observe that **MF achieves accuracy very close to the batch RF versions** (Breiman-RF, ERT-$k$, ERT-1) trained on the same fraction of the data. **MF significantly outperforms ORF-Saffari trained on the same fraction of training data**. In batch RF versions, the same training data can be used to evaluate candidate splits at a node and its children. However, in the online RF versions (ORF-Saffari and ORF-Denil), incoming training examples are used to evaluate candidate splits just at a current leaf node and new training data are required to evaluate candidate splits every time a new leaf node is created. Saffari et al. [20] recommend multiple passes through the training data to increase the effective number of training samples. In a realistic streaming data setup, where training examples cannot be stored for multiple passes, MF would require significantly fewer examples than ORF-Saffari to achieve the same accuracy.

Comparing test accuracy vs training time on *usps*, *satimages* and *letter* datasets, we observe that **MF is at least an order of magnitude faster than re-trained batch versions and ORF-Saffari**. For ORF-Saffari, we plot test accuracy at the end of every additional pass; hence it contains additional markers compared to the top row which plots results after a single pass. Re-training batch RF using 100 mini-batches is unfair to MF; in a streaming data setup where the model is updated when a new training instance arrives, MF would be significantly faster than the re-trained batch versions.

Assuming trees are balanced after adding each data point, it can be shown that computational cost of MF scales as $\mathcal{O}(N \log N)$ whereas that of re-trained batch RF scales as $\mathcal{O}(N^2 \log N)$ (Appendix C). Appendix E shows that the average depth of the forests trained on above datasets scales as $\mathcal{O}(\log N)$.

It is remarkable that choosing splits independent of labels achieves competitive classification performance. This phenomenon has been observed by others as well—for example, Cutler and Zhao [7] demonstrate that their PERT classifier (which is similar to batch version of MF) achieves test accuracy comparable to Breiman-RF on many real world datasets. However, in the presence of irrelevant features, methods which choose splits independent of labels (MF, ERT-1) perform worse than Breiman-RF and ERT-$k$ (but still better than ORF-Saffari) as indicated by the results on the *dna* dataset. We trained MF and ERT-1 using just the most relevant 60 attributes amongst the 180 attributes[5]—these results are indicated as MF$^\dagger$ and ERT-1$^\dagger$ in Figure 3. We observe that, as expected, filtering out irrelevant features significantly improves performance of MF and ERT-1.

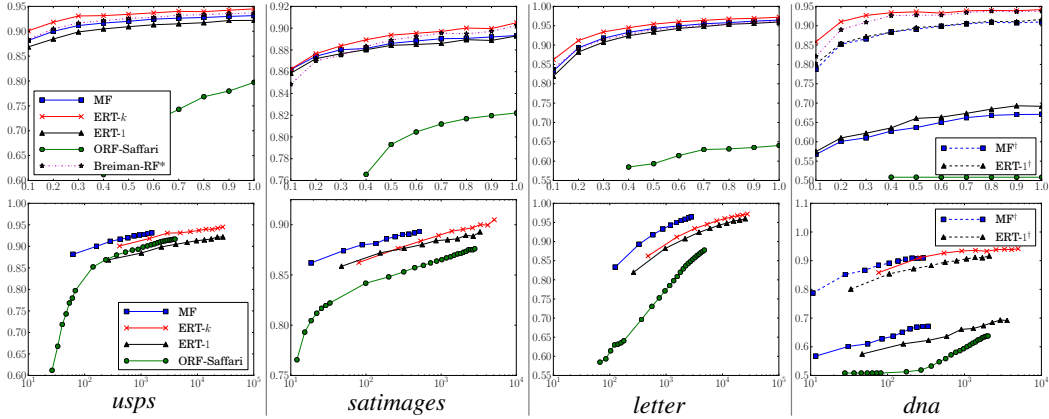

**Figure 3:** Results on various datasets: $y$-axis is test accuracy in both rows. $x$-axis is fraction of training data for the top row and training time (in seconds) for the bottom row. We used the pre-defined train/test split. For *usps* dataset $D = 256, K = 10, N_{\text{train}} = 7291, N_{\text{test}} = 2007$; for *satimages* dataset $D = 36, K = 6, N_{\text{train}} = 3104, N_{\text{test}} = 2000$; *letter* dataset $D = 16, K = 26, N_{\text{train}} = 15000, N_{\text{test}} = 5000$; for *dna* dataset $D = 180, K = 3, N_{\text{train}} = 1400, N_{\text{test}} = 1186$.

# 8 Discussion

We have introduced *Mondrian forests*, a novel class of random forests, which can be trained incrementally in an efficient manner. MF significantly outperforms existing online random forests in terms of training time as well as number of training instances required to achieve a particular test accuracy. Remarkably, MF achieves competitive test accuracy to batch random forests trained on the same fraction of the data. MF is unable to handle lots of irrelevant features (since splits are chosen independent of the labels)—one way to use labels to guide splits is via recently proposed Sequential Monte Carlo algorithm for decision trees [14]. The computational complexity of MF is linear in the number of dimensions (since rectangles are represented explicitly) which could be expensive for high dimensional data; we will address this limitation in future work. Random forests have been tremendously influential in machine learning for a variety of tasks; hence lots of other interesting extensions of this work are possible, e.g. MF for regression, theoretical bias-variance analysis of MF, extensions of MF that use hyperplane splits instead of axis-aligned splits.

### Acknowledgments

We would like to thank Charles Blundell, Gintare Dziugaite, Creighton Heaukulani, José Miguel Hernández-Lobato, Maria Lomeli, Alex Smola, Heiko Strathmann and Srini Turaga for helpful discussions and feedback on drafts. BL gratefully acknowledges generous funding from the Gatsby Charitable Foundation. This research was carried out in part while DMR held a Research Fellowship at Emmanuel College, Cambridge, with funding also from a Newton International Fellowship through the Royal Society. YWT's research leading to these results was funded in part by the European Research Council under the European Union's Seventh Framework Programme (FP7/2007-2013) ERC grant agreement no. 617411.

## Footnotes

[1]Roy and Teh [19] studied the distribution of $\{\mathcal{M}_t : t \leq \lambda\}$ and refered to $\lambda$ as the *budget*. See [18, Chp. 5] for more details. We will refer to $t$ as time, not be confused with discrete time in the online learning setting.

[2]Taking the discount parameter to zero leads to a Dirichlet process . Hierarchies of NSPs admit more tractable approximations than hierarchies of Dirichlet processes [24], hence our choice here.

[3]The *scikit-learn* implementation uses highly optimized C code, hence we do not compare our runtimes with the *scikit-learn* implementation. The ERT implementation in *scikit-learn* achieves very similar test accuracy as our ERT implementation, hence we do not report those results here.

[4]https://archive.ics.uci.edu/ml/machine-learning-databases/mushroom/agaricus-lepiota.names

[5]https://www.sgi.com/tech/mlc/db/DNA.names

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
