[Supplementary Material]

# Mondrian Forests: Efficient Online Random Forests
## Supplementary Material

## A  Posterior inference and prediction using the HNSP

Recall that we use a hierarchical Bayesian approach to specify a smooth label distribution $p_T(y|\boldsymbol{x}, \mathcal{D}_{1:N})$ for each tree $T$. The label prediction at a test point $\boldsymbol{x}$ will depend on where $\boldsymbol{x}$ falls relative to the existing data in the tree $T$. In this section, we assume that $\boldsymbol{x}$ lies within one of the leaf nodes in $T$, i.e., $\boldsymbol{x} \in B^x_{\mathsf{leaf}(\boldsymbol{x})}$, where $\mathsf{leaf}(\boldsymbol{x}) \in \mathsf{leaves}(\mathsf{T})$. If $\boldsymbol{x}$ does not lie within any of the leaf nodes in $T$, i.e., $\boldsymbol{x} \notin \cup_{j \in \mathsf{leaves}(\mathsf{T})} B^x_j$, one could extend the tree by sampling $T'$ from $\mathrm{MTx}(\lambda, T, \boldsymbol{x})$, such that $\boldsymbol{x}$ lies within a leaf node in $T'$ and apply the procedure described below using the extended tree $T'$. Appendix B describes this case in more detail.

Given training data $\mathcal{D}_{1:N}$, a Mondrian tree $T$ and the hierarchical prior over $\mathcal{G}$, the predictive label distribution $p_T(y|\boldsymbol{x}, \mathcal{D}_{1:N})$ is obtained by integrating over $\mathcal{G}$, i.e.

$$p_T(y|\boldsymbol{x}, \mathcal{D}_{1:N}) = \mathbb{E}_{\mathcal{G} \sim p_T(\mathcal{G}|\mathcal{D}_{1:N})}[G_{\mathsf{leaf}(\boldsymbol{x}),y}] = \overline{G}_{\mathsf{leaf}(\boldsymbol{x}),y}.$$

Hence, the prediction is given by $\overline{G}_{\mathsf{leaf}(\boldsymbol{x})}$, the posterior mean at $\mathsf{leaf}(\boldsymbol{x})$. The posterior mean $\overline{G}_{\mathsf{leaf}(\boldsymbol{x})}$ can be computed using existing techniques, which we review in the rest of this section.

Posterior inference in the HNSP is a special case of posterior inference in hierarchical PYP (HPYP). Teh [22] considers the HPYP with multinomial likelihood (in the context of language modeling)—the model considered here (HNSP with multinomial likelihood) is a special case of [22]. Hence, we just sketch the high level picture and refer the reader to [22] for further details. We first describe posterior inference given $N$ data points $\mathcal{D}_{1:N}$ (batch setting), and later explain how to adapt inference to the online setting. Finally, we describe the computation of the predictive posterior distribution.

### Batch setting

Posterior inference is done using the Chinese restaurant process representation, wherein every node of the decision tree is a restaurant; the training data points are the customers seated in the tables associated with the leaf node restaurants; these tables are in turn customers at the tables in their corresponding parent level restaurant; the dish served at each table is the class label. Exact inference is intractable and hence we resort to approximations. In particular, we use the approximation known as the interpolated Kneser-Ney (IKN) smoothing, a popular smoothing technique for language modeling [13]. The IKN smoothing can be interpreted as an approximate inference scheme for the HPYP, where the number of tables serving a particular dish in a restaurant is at most one [22]. More precisely, if $c_{j,k}$ denotes the number of customers at restaurant $j$ eating dish $k$ and $\mathsf{tab}_{j,k}$ denotes the number of tables at restaurant $j$ serving dish $k$, the IKN approximation sets $\mathsf{tab}_{j,k} = \min(c_{j,k}, 1)$. The counts $c_{j,k}$ and $\mathsf{tab}_{j,k}$ can be computed in a single bottom-up pass as follows: for every leaf node $j \in \mathsf{leaves}(\mathsf{T})$, $c_{j,k}$ is simply the number of training data points with label $k$ at node $j$; for every internal node $j \in \mathsf{T} \setminus \mathsf{leaves}(\mathsf{T})$, we set $c_{j,k} = \mathsf{tab}_{\mathsf{left}(j),k} + \mathsf{tab}_{\mathsf{right}(j),k}$. For a leaf node $j$, this procedure is summarized in Algorithm 5. (Note that this pseudocode just serves as a reference; in practice, these counts are updated in an online fashion, as described in Algorithm 6.)

**Algorithm 5** InitializePosteriorCounts($j$)

1: For all $k$, set $c_{jk} = \#\{n \in N(j) : y_n = k\}$
2: Initialize $j' = j$
3: **while** True **do**
4:     **if** $j' \notin \mathsf{leaves}(\mathsf{T})$ **then**
5:         For all $k$, set $c_{j'k} = \mathsf{tab}_{\mathsf{left}(j'),k} + \mathsf{tab}_{\mathsf{right}(j'),k}$
6:     For all $k$, set $\mathsf{tab}_{j'k} = \min(c_{j'k}, 1)$                ▷ *IKN approximation*
7:     **if** $j' = \epsilon$ **then**
8:         **return**
9:     **else**
10:         $j' \leftarrow \mathsf{parent}(j')$

**Posterior inference: online setting**

It is straightforward to extend inference to the online setting. Adding a new data point $\mathcal{D} = (\boldsymbol{x}, y)$ affects only the counts along the path from the root to the leaf node of that data point. We update the counts in a bottom-up fashion, starting at the leaf node containing the data point, $\mathsf{leaf}(\boldsymbol{x})$. Due to the nature of the IKN approximation, we can stop at the internal node $j$ where $c_{j,y} = 1$ and need not traverse up till the root. This procedure is summarized in Algorithm 6.

**Algorithm 6** $\mathsf{UpdatePosteriorCounts}(j, y)$

1: $\quad c_{jy} \leftarrow c_{jy} + 1$
2: $\quad$ Initialize $j' = j$
3: $\quad$ **while** True **do**
4: $\qquad$ **if** $\mathsf{tab}_{j'y} = 1$ **then** $\qquad\qquad\qquad\qquad\qquad\quad$ ▷ *none of the counts above need to be updated*
5: $\qquad\quad$ **return**
6: $\qquad$ **else**
7: $\qquad\quad$ **if** $j' \notin \mathsf{leaves}(\mathsf{T})$ **then**
8: $\qquad\qquad$ $c_{j'y} = \mathsf{tab}_{\mathsf{left}(j'),y} + \mathsf{tab}_{\mathsf{right}(j'),y}$
9: $\qquad\quad$ $\mathsf{tab}_{j'y} = \min(c_{j'y}, 1)$ $\qquad\qquad\qquad\qquad\qquad\qquad$ ▷ *IKN approximation*
10: $\qquad\quad$ **if** $j' = \epsilon$ **then**
11: $\qquad\qquad$ **return**
12: $\qquad\quad$ **else**
13: $\qquad\qquad$ $j' \leftarrow \mathsf{parent}(j')$

**Predictive posterior computation**   Given the counts $c_{j,k}$ and table assignments $\mathsf{tab}_{j,k}$, the predictive probability (i.e., posterior mean) at node $j$ can be computed recursively as follows:

$$\overline{G}_{jk} = \begin{cases} \dfrac{c_{j,k} - d_j \mathsf{tab}_{j,k}}{c_{j,\cdot}} + \dfrac{d_j \mathsf{tab}_{j,\cdot}}{c_{j,\cdot}} \overline{G}_{\mathsf{parent}(j),k} & c_{j,\cdot} > 0, \\ \overline{G}_{\mathsf{parent}(j),k} & c_{j,\cdot} = 0, \end{cases} \tag{4}$$

where $c_{j,\cdot} = \sum_k c_{j,k}$, $\mathsf{tab}_{j,\cdot} = \sum_k \mathsf{tab}_{j,k}$, and $d_j := \exp\big(-\gamma(\tau_j - \tau_{\mathsf{parent}(j)})\big)$ is the *discount* for node $j$, defined in Section 4. Informally, the discount interpolates between the counts $c$ and the prior. If the discount $d_j \approx 1$, then $\overline{G}_j$ is more like its parent $\overline{G}_{\mathsf{parent}(j)}$. If $d_j \approx 0$, then $\overline{G}_j$ weights the counts more. These predictive probabilities can be computed in a single top-down pass as shown in Algorithm 7.

**Algorithm 7** $\mathsf{ComputePosteriorPredictiveDistribution}(T, \mathcal{G})$

1: $\quad$ ▷ *Description of top-down pass to compute posterior predictive distribution given by (4)*
2: $\quad$ ▷ $\overline{G}_{jk}$ *denotes the posterior probability of* $y = k$ *at node* $j$
3: $\quad$ Initialize the ordered set $J = \{\epsilon\}$
4: $\quad$ **while** $J$ not empty **do**
5: $\qquad$ Pop the first element of $J$
6: $\qquad$ **if** $j = \epsilon$ **then**
7: $\qquad\quad$ $\overline{G}_{\mathsf{parent}(\epsilon)} = H$
8: $\qquad$ Set $d = \exp\big(-\gamma(\tau_j - \tau_{\mathsf{parent}(j)})\big)$
9: $\qquad$ For all $k$, set $\overline{G}_{jk} = c_{j,\cdot}^{-1}\Big( c_{j,k} - d\,\mathsf{tab}_{j,k} + d\,\mathsf{tab}_{j,\cdot}\,\overline{G}_{\mathsf{parent}(j),k} \Big)$
10: $\qquad$ **if** $j \notin \mathsf{leaves}(\mathsf{T})$ **then**
11: $\qquad\quad$ Append $\mathsf{left}(j)$ and $\mathsf{right}(j)$ to the end of the ordered set $J$

## B   Prediction using Mondrian tree

Let $\boldsymbol{x}$ denote a test data point. We are interested in the predictive probability of $y$ at $\boldsymbol{x}$, denoted by $p_T(y|\boldsymbol{x}, \mathcal{D}_{1:N})$. As in typical decision trees, the process involves a top-down tree traversal, starting from the root. If $\boldsymbol{x}$ is already 'contained' in the tree $T$, i.e., if $\boldsymbol{x} \in B_j^x$ for some leaf $j \in \mathsf{leaves}(\mathsf{T})$, then the prediction is taken to be $\overline{G}_{\mathsf{leaf}(\boldsymbol{x})}$, which is computed as described in Appendix A. Otherwise,

we somehow need to incorporate $\boldsymbol{x}$. One choice is to extend $T$ by sampling $T'$ from $\mathsf{MTx}(\lambda, T, \boldsymbol{x})$ as described in Algorithm 3, and set the prediction to $\overline{G}_j$, where $j \in \mathsf{leaves}(\mathsf{T}')$ is the leaf node containing $\boldsymbol{x}$. A particular extension $T'$ might lead to an overly confident prediction; hence, we average over *every* possible extension $T'$. This expectation can be carried out analytically, using properties of the Mondrian process, as we show below.

Let $\mathsf{ancestors}(j)$ denote the set of all ancestors of node $j$. Let $\mathsf{path}(j) = \{j\} \cup \mathsf{ancestors}(j)$, that is, the set of all nodes along the ancestral path from $j$ to the root. Recall that $\mathsf{leaf}(\boldsymbol{x})$ is the unique leaf node in $\mathsf{T}$ such that $\boldsymbol{x} \in B_{\mathsf{leaf}(\boldsymbol{x})}$. If the test point $\boldsymbol{x} \in B_{\mathsf{leaf}(\boldsymbol{x})}^x$ (i.e., $\boldsymbol{x}$ lies within the 'gray rectangle' at the leaf node), it can never branch off; else, it can branch off at one or more points along the path from the root to $\mathsf{leaf}(\boldsymbol{x})$. More precisely, if $\boldsymbol{x}$ lies outside $B_j^x$ at node $j$, the probability that $\boldsymbol{x}$ will branch off into its own node at node $j$, denoted by[6] $p_j^s(\boldsymbol{x})$, is equal to the probability that a split exists in $B_j$ outside $B_j^x$, which is

$$ p_j^s(\boldsymbol{x}) = 1 - \exp\bigl(-\Delta_j \eta_j(\boldsymbol{x})\bigr), \quad \text{where } \eta_j(\boldsymbol{x}) = \sum_d \bigl(\mathsf{max}(x_d - u_{jd}^x, 0) + \mathsf{max}(\ell_{jd}^x - x_d, 0)\bigr), $$

and $\Delta_j = \tau_j - \tau_{\mathsf{parent}(j)}$. Note that $p_j^s(\boldsymbol{x}) = 0$ if $\boldsymbol{x}$ lies within $B_j^x$ (i.e., if $\ell_{jd}^x \leq x_d \leq u_{jd}^x$ for all $d$). The probability of $\boldsymbol{x}$ not branching off before reaching node $j$ is given by $\prod_{j' \in \mathsf{ancestors}(j)} (1 - p_{j'}^s(\boldsymbol{x}))$.

If $\boldsymbol{x} \in B_{\mathsf{leaf}(\boldsymbol{x})}^x$, the prediction is given by $\overline{G}_{\mathsf{leaf}(\boldsymbol{x})}$. If there is a split in $B_j$ outside $B_j^x$, let $\tilde{j}$ denote the new parent of $j$ and $\mathsf{child}(\tilde{j})$ denote the child node containing just the test data point,; in this case, the prediction is $\overline{G}_{\mathsf{child}(\tilde{j})}$. Averaging over the location where the test point branches off, we obtain

$$ p_T(y|\boldsymbol{x}, \mathcal{D}_{1:N}) = \sum_{j \in \mathsf{path}(\mathsf{leaf}(\boldsymbol{x}))} \Bigl( \prod_{j' \in \mathsf{ancestors}(j)} (1 - p_{j'}^s(\boldsymbol{x})) \Bigr) F_j(\boldsymbol{x}), \tag{5} $$

where

$$ F_j(\boldsymbol{x}) = p_j^s(\boldsymbol{x}) \mathbb{E}_{\Delta_{\tilde{j}}} \Bigl[ \overline{G}_{\mathsf{child}(\tilde{j})} \Bigr] + \mathbb{1}[j = \mathsf{leaf}(\boldsymbol{x})](1 - p_j^s(\boldsymbol{x})) \overline{G}_{\mathsf{leaf}(\boldsymbol{x})}. \tag{6} $$

The second term in $F_j(\boldsymbol{x})$ needs to be computed only for the leaf node $\mathsf{leaf}(\boldsymbol{x})$ and is simply the posterior mean of $G_{\mathsf{leaf}(\boldsymbol{x})}$ weighted by $1 - p_{\mathsf{leaf}(\boldsymbol{x})}^s(\boldsymbol{x})$. The posterior mean of $G_{\mathsf{leaf}(x)}$, given by $\overline{G}_{\mathsf{leaf}(\boldsymbol{x})}$, can be computed using (4). The first term in $F_j(\boldsymbol{x})$ is simply the posterior mean of $G_{\mathsf{child}(\tilde{j})}$, averaged over $\Delta_{\tilde{j}}$, weighted by $p_j^s(\boldsymbol{x})$. Since no labels are observed in $\mathsf{child}(\tilde{j})$, $c_{\mathsf{child}(\tilde{j}),\cdot} = 0$, hence from (4), we have $\overline{G}_{\mathsf{child}(\tilde{j})} = \overline{G}_{\tilde{j}}$. We compute $\overline{G}_{\tilde{j}}$ using (4). We average over $\Delta_{\tilde{j}}$ due to the fact that the discount in (4) for the node $\tilde{j}$ depends on $\tau_{\tilde{j}} - \tau_{\mathsf{parent}(\tilde{j})} = \Delta_{\tilde{j}}$. To average over all valid split times $\tau_{\tilde{j}}$, we compute expectation w.r.t. $\Delta_{\tilde{j}}$ which is distributed according to a truncated exponential with rate $\eta_j(\boldsymbol{x})$, truncated to the interval $[0, \Delta_j]$.

The procedure for computing $p_T(y|\boldsymbol{x}, \mathcal{D}_{1:N})$ for any $\boldsymbol{x} \in \mathbb{R}^D$ is summarized in Algorithm 8. The predictive probability assigned by a Mondrian forest is the average of the predictive probability of the $M$ trees, i.e., $\frac{1}{M} \sum_m p_{T_m}(y|\boldsymbol{x}, \mathcal{D}_{1:N})$.

## C  Computational complexity

We discuss the computational complexity associated with a single Mondrian tree. The complexity of a forest is simply $M$ times that of a single tree; however, this computation can be trivially parallelized since there is no interaction between the trees. Assume that the $N$ data points are processed one by one. Assuming the data points form a balanced binary tree after each update, the computational cost of processing the $n^{th}$ data point is at most $\mathcal{O}(\log n)$ (add the data point into its own leaf, update posterior counts for HNSP in bottom-up pass from leaf to root). The overall cost to process $N$ data points is $\mathcal{O}(\sum_{n=1}^N \log n) = \mathcal{O}(\log N!)$, which for large $N$ tends to $\mathcal{O}(N \log N)$ (using Stirling approximation for the factorial function). For offline RF and ERT, the expected complexity with $n$ data points is $\mathcal{O}(n \log n)$. The complexity of the re-trained version is $\mathcal{O}(\sum_{n=1}^N n \log n) = \mathcal{O}(\log \prod_{n=1}^N n^n)$, which for large $N$ tends to $\mathcal{O}(N^2 \log N)$ (using asymptotic expansion of the hyper factorial function).

**Algorithm 8** Predict$(T, \boldsymbol{x})$

1: ▷ *Description of prediction using a Mondrian tree, given by ([5](#))*
2: Initialize $j = \epsilon$ and $p_{\mathsf{NotSeparatedYet}} = 1$
3: Initialize $\mathbf{s} = \mathbf{0}_K$         ▷ $\mathbf{s}$ *is $K$-dimensional vector where $s_k = p_T(y = k | \boldsymbol{x}, \mathcal{D}_{1:N})$*
4: **while** True **do**
5:      Set $\Delta_j = \tau_j - \tau_{\mathsf{parent}(j)}$ and $\eta_j(\boldsymbol{x}) = \sum_d \big( \max(x_d - u_{jd}^x, 0) + \max(\ell_{jd}^x - x_d, 0) \big)$
6:      Set $p_j^s(\boldsymbol{x}) = 1 - \exp\big(-\Delta_j \eta_j(\boldsymbol{x})\big)$
7:      **if** $p_j^s(\boldsymbol{x}) > 0$ **then**
8:          ▷ *Let $\boldsymbol{x}$ branch off into its own node* child$(\tilde{j})$, *creating a new node $\tilde{j}$ which is the parent of $j$ and* child$(\tilde{j})$. $\overline{G}_{\mathsf{child}(\tilde{j})} = \overline{G}_{\tilde{j}}$ *from ([4](#)) since $c_{\mathsf{child}(\tilde{j}),\cdot} = 0$.*
9:          Compute expected discount $\bar{d} = \mathbb{E}_{\Delta}[\exp(-\gamma \Delta)]$ where $\Delta$ is drawn from a truncated exponential with rate $\eta_j(\boldsymbol{x})$, truncated to the interval $[0, \Delta_j]$.
10:          For all $k$, set $c_{\tilde{j},k} = \mathsf{tab}_{\tilde{j},k} = \min(c_{j,k}, 1)$
11:          For all $k$, set $\overline{G}_{\tilde{j}k} = c_{\tilde{j},\cdot}^{-1}\Big(c_{\tilde{j},k} - \bar{d}\,\mathsf{tab}_{\tilde{j},k} + \bar{d}\,\mathsf{tab}_{\tilde{j},\cdot}\,\overline{G}_{\mathsf{parent}(\tilde{j}),k}\Big)$   ▷ *Algorithm [7](#) and ([6](#))*
12:          For all $k$, update $s_k \leftarrow s_k + p_{\mathsf{NotSeparatedYet}}\, p_j^s(\boldsymbol{x})\overline{G}_{\tilde{j}k}$
13:      **if** $j \in$ leaves$(\mathsf{T})$ **then**
14:          For all $k$, update $s_k \leftarrow s_k + p_{\mathsf{NotSeparatedYet}}(1 - p_j^s(\boldsymbol{x}))\overline{G}_{jk}$       ▷ *Algorithm [7](#) and ([6](#))*
15:          **return** predictive probability $\mathbf{s}$ where $s_k = p_T(y = k | \boldsymbol{x}, \mathcal{D}_{1:N})$
16:      **else**
17:          $p_{\mathsf{NotSeparatedYet}} \leftarrow p_{\mathsf{NotSeparatedYet}}(1 - p_j^s(\boldsymbol{x}))$
18:          **if** $x_{\delta_j} \leq \xi_j$ **then** $j \leftarrow$ left$(j)$ **else** $j \leftarrow$ right$(j)$      ▷ *recurse to the child where $\boldsymbol{x}$ lies*

# D    Pseudocode for paused Mondrians

In this section, we discuss versions of SampleMondrianBlock and ExtendMondrianBlock for paused Mondrians. For completeness, we also provide the updates necessary for the IKN approximation.

**Algorithm 9** SampleMondrianBlock$\big(j, \mathcal{D}_{N(j)}, \lambda\big)$ version that depends on labels

1: Add $j$ to $\mathsf{T}$
2: For all $d$, set $\ell_{jd}^x = \min(\boldsymbol{X}_{N(j),d}), u_{jd}^x = \max(\boldsymbol{X}_{N(j),d})$     ▷ *dimension-wise* min *and* max
3: **if** AllLabelsIdentical$(Y_{N(j)})$ **then**
4:      Set $\tau_j = \lambda$                                 ▷ *pause Mondrian*
5: **else**
6:      Sample $E$ from exponential distribution with rate $\sum_d (u_{jd}^x - \ell_{jd}^x)$
7:      Set $\tau_j = \tau_{\mathsf{parent}(j)} + E$
8: **if** $\tau_j < \lambda$ **then**
9:      Sample split dimension $\delta_j$ with probability of choosing $d$ proportional to $u_{jd}^x - \ell_{jd}^x$
10:      Sample split location $\xi_j$ along dimension $\delta_j$ from an uniform distribution over $\mathcal{U}[\ell_{jd}^x, u_{jd}^x]$
11:      Set $N(\mathsf{left}(j)) = \{n \in N(j) : \boldsymbol{X}_{n,\delta_j} \leq \xi_j\}$ and $N(\mathsf{right}(j)) = \{n \in N(j) : \boldsymbol{X}_{n,\delta_j} > \xi_j\}$
12:      SampleMondrianBlock$\big(\mathsf{left}(j), \mathcal{D}_{N(\mathsf{left}(j))}, \lambda\big)$
13:      SampleMondrianBlock$\big(\mathsf{right}(j), \mathcal{D}_{N(\mathsf{right}(j))}, \lambda\big)$
14: **else**
15:      Set $\tau_j = \lambda$ and add $j$ to leaves$(\mathsf{T})$             ▷ *$j$ is a leaf node*
16:      InitializePosteriorCounts$(j)$                   ▷ *Algorithm [5](#)*

**Algorithm 10** ExtendMondrianBlock$(T, \lambda, j, \mathcal{D})$ version that depends on labels

1: **if** AllLabelsIdentical$(Y_{N(j)})$ **then** ▷ *paused Mondrian leaf*
2:     Update extent $\boldsymbol{\ell}_j^x \leftarrow \min(\boldsymbol{\ell}_j^x, \boldsymbol{x}), \mathbf{u}_j^x \leftarrow \max(\mathbf{u}_j^x, \boldsymbol{x})$
3:     Append $\mathcal{D}$ to $\mathcal{D}_{N(j)}$ ▷ *append $\boldsymbol{x}$ to $X_{N(j)}$ and $y$ to $Y_{N(j)}$*
4:     **if** $y = \mathsf{unique}(Y_{N(j)})$ **then**
5:         UpdatePosteriorCounts$(j, y)$ ▷ *Algorithm 6*
6:         **return** ▷ *continue pausing*
7:     **else**
8:         Remove $j$ from leaves$(\mathsf{T})$
9:         SampleMondrianBlock$(j, \mathcal{D}_{N(j)}, \lambda)$ ▷ *un-pause Mondrian*
10: **else**
11:     Set $\mathbf{e}^\ell = \max(\boldsymbol{\ell}_j^x - \boldsymbol{x}, 0)$ and $\mathbf{e}^u = \max(\boldsymbol{x} - \mathbf{u}_j^x, 0)$ ▷ $\mathbf{e}^\ell = \mathbf{e}^u = \mathbf{0}_D$ *if* $\boldsymbol{x} \in B_j^x$
12:     Sample $E$ from exponential distribution with rate $\sum_d (e_d^\ell + e_d^u)$
13:     **if** $\tau_{\mathsf{parent}(j)} + E < \tau_j$ **then** ▷ *introduce new parent for node $j$*
14:         Create new Mondrian block $\tilde{j}$ where $\boldsymbol{\ell}_{\tilde{j}}^x = \min(\boldsymbol{\ell}_j^x, \boldsymbol{x})$ and $\mathbf{u}_{\tilde{j}}^x = \max(\mathbf{u}_j^x, \boldsymbol{x})$
15:         Sample $\delta_{\tilde{j}}$ with $\Pr(\delta_{\tilde{j}} = d)$ proportional to $e_d^\ell + e_d^u$
16:         **if** $x_{\delta_{\tilde{j}}} > u_{j,\delta_{\tilde{j}}}^x$, **then** sample $\xi_{\tilde{j}}$ from $\mathcal{U}[u_{j,\delta_{\tilde{j}}}^x, x_{\delta_{\tilde{j}}}]$, **else** sample $\xi_{\tilde{j}}$ from $\mathcal{U}([x_{\delta_{\tilde{j}}}, \ell_{j,\delta_{\tilde{j}}}^x])$
17:         **if** $j = \epsilon$ **then** ▷ *set $\tilde{j}$ as the new root*
18:             $\epsilon \leftarrow \tilde{j}$
19:         **else** ▷ *set $\tilde{j}$ as child of* parent$(j)$
20:             **if** $j = \mathsf{left}(\mathsf{parent}(j))$, **then** $\mathsf{left}(\mathsf{parent}(j)) \leftarrow \tilde{j}$, **else** $\mathsf{right}(\mathsf{parent}(j)) \leftarrow \tilde{j}$
21:         **if** $x_{\delta_{\tilde{j}}} > \xi_{\tilde{j}}$ **then**
22:             Set $\mathsf{left}(\tilde{j}) = j$ and SampleMondrianBlock$(\mathsf{right}(\tilde{j}), \mathcal{D}, \lambda)$ ▷ *create new leaf for $x$*
23:         **else**
24:             Set $\mathsf{right}(\tilde{j}) = j$ and SampleMondrianBlock$(\mathsf{left}(\tilde{j}), \mathcal{D}, \lambda)$ ▷ *create new leaf for $x$*
25:     **else**
26:         Update $\boldsymbol{\ell}_j^x \leftarrow \min(\boldsymbol{\ell}_j^x, \boldsymbol{x}), \mathbf{u}_j^x \leftarrow \max(\mathbf{u}_j^x, \boldsymbol{x})$ ▷ *update extent of node $j$*
27:         **if** $j \notin \mathsf{leaves}(\mathsf{T})$ **then** ▷ *return if $j$ is a leaf node, else recurse down the tree*
28:             **if** $x_{\delta_j} \leq \xi_j$ **then** child$(j) = \mathsf{left}(j)$ **else** child$(j) = \mathsf{right}(j)$
29:             ExtendMondrianBlock$(T, \lambda, \mathsf{child}(j), \mathcal{D})$ ▷ *recurse on child containing $x$*

## E   Depth of trees

We computed the average depth of the trees in the forest, where depth of a leaf node is weighted by fraction of data points at that leaf node. The hyper-parameter settings and experimental setup are described in Section 7. Table 1 reports the average depth (and standard deviations) for Mondrian forests trained on different datasets. The values suggest that the depth of the forest scales as $\log N$ rather than $N$.

| Dataset | $N_{\mathsf{train}}$ | $\log_2 N_{\mathsf{train}}$ | depth |
|---------|------------|--------------------|-------------|
| *usps* | 7291 | 12.8 | $19.1 \pm 1.3$ |
| *satimages* | 3104 | 11.6 | $17.4 \pm 1.6$ |
| *letter* | 15000 | 13.9 | $23.2 \pm 1.8$ |
| *dna* | 1400 | 10.5 | $12.0 \pm 0.3$ |

**Table 1:** Average depth of Mondrian forests trained on different datasets.

## F   Comparison to dynamic trees

Dynamic trees [21] approximate the Bayesian posterior over decision trees in an online fashion. Specifically, dynamic trees maintain a particle approximation to the true posterior; the prediction at a test point is a weighted average of the predictions made by the individual particles. While this averaging procedure appears similar to online random forests at first sight, there is a key difference: MF (and other random forests) performs ensemble model combination whereas dynamic trees use Bayesian model averaging. In the limit of infinite data, the Bayesian posterior would converge to a

single tree [15], whereas MF would still average predictions over multiple trees. Hence, we expect MF to outperform dynamic trees in scenarios where a single decision tree is insufficient to explain the data.

To experimentally validate our hypothesis, we evaluate the empirical performance of dynamic trees using the dynaTree[7] R package provided by the authors of the paper. Note that while dynamic trees can use 'linear leaves' (strong since prediction at a leaf depends on X) or 'constant leaves' for regression tasks, they use 'multinomial leaves' for classification tasks which corresponds to a 'weak learner'. We set the number of particles to 100 (equals the number of trees used in MF) and the number of passes, $R = 2$ (their code does not support $R = 1$) and set the remaining parameters to their default values. Fig. 4 compares the performance of dynamic trees to MF and other random forest variants. (The performance of all methods other than dynamic trees is identical to that of Fig. 3.)

**Figure 4:** Results on various datasets: $y$-axis is test accuracy in both rows. $x$-axis is fraction of training data. The setup is identical to that of Fig. 3. MF achieves significantly higher test accuracies than dynamic trees on *usps*, *satimages* and *letter* datasets and MF$^\dagger$ achieves similar test accuracy as dynamic trees on the *dna* dataset.

We observe that MF achieves significantly higher test accuracies than dynamic trees on *usps*, *satimages* and *letter* datasets. On *dna* dataset, dynamic trees outperform MF (indicating the usefulness of using labels to guide splits) — however, MF with feature selection (MF$^\dagger$) achieves similar performance as dynamic trees. All the batch random forest methods are superior to dynamic trees which suggests that decision trees are not sufficient to explain these real world datasets and that model combination is helpful.

## Footnotes

[6]The superscript $s$ in $p_j^s(\boldsymbol{x})$ is used to denote the fact that this split 'separates' the test data point $\boldsymbol{x}$ into its own leaf node.

[7]http://cran.r-project.org/web/packages/dynaTree/index.html