[Reviews · NeurIPS 2014]

Submitted by Assigned_Reviewer_2

The paper describes a Bayesian model for online learning in the context of random forests models for supervised classification. The main contribution of the paper is the formulation of a novel prior on binary rooted trees that relies on the Mondrian process. An additional novelty of the paper is the use of hierarchical normalized stable processes as priors for the probabilities of the different classes at each terminal node.

The paper is well written and the formulation novel. A very interesting feature of the model is that is consistent over marginalization (the authors call this property projectivity), which is particularly important in online settings.

General comments: I just have a couple of minor comments:

1) In lines 67 and 68 the authors acknowledge that “Eq. (2) is ensemble model combination, not Bayesian model averaging”. At least from the point of view of mean squared prediction errors, ensemble combinations with equal weights are suboptimal. I assume that the reason to proceed in this way is to improve computational speed (basically, to avoid computing marginal likelihoods). I suspect that “this is what everybody else does” is another reason. However, a brief discussion would be appropriate.

2) I think that dismissing dynamic decision trees such as the ones discussed in [16] because “random forests are known to outperform single decision trees” (lines 341 and 342) is not appropriate in this case. For example, [16] uses sophisticated models at each tip (“strong learners”), while most comparisons between random forests and random trees that I am familiar with assume that “weak learners” are used at the tips at the tips of the trees in both cases. Including them in the evaluations of Section
Summary: The paper is well written, the topic interesting, and the model novel. I think that some aspects of the evaluation could be improved, but overall this is a strong paper.

Submitted by Assigned_Reviewer_15

The paper presents an efficient online algorithm called Mondrian forest. A nice characteristic of this model is that the distribution of online Mondrian forest is the same as its batch counterpart.

To the best of my knowledge this is original and innovative work. Concerning clarity, the introduction needs to be improved. I would move section 6 (related works) to introduction.

It is an interesting idea. My only concern is that the partition is independent of the response variable, and this may lead to poor performance as the number of features increases. The model has been tested on low dimensional spaces (small D). It may be useful to see how the model performs as D increases.

Summary: Efficient online learning algorithm based on Mondrian process. The work is original and innovative but I am concerned about its performance on higher dimensional spaces.

Submitted by Assigned_Reviewer_36

In this paper, the authors introduce Mondrian forests, a classifier inspired by random forests, Mondrian processes, and interpolated Kneser-Ney (IKN). They define Mondrian trees (MT), show how a forest is built from such trees, discuss relations to Mondrian processes and hierarchical Pitman-Yor processes (HPYP). They describe how online training and prediction work. Finally, they show that Mondrian forests provide competitive predictive accuracy as a function of the amount of training data seen and some of the best predictive accuracy as a function of amount of time spent. They discuss briefly how this accuracy breaks down when too many useless features are input and how future work might ameliorate this issue.

Overall the model in this paper is interesting, the experiments are convincing, and the application to streaming data in particular seems timely and important. I definitely would like to see this research published.

My major qualm with this manuscript is that it reads as though the authors have tried to fit a journal paper into the space allotted by NIPS. As it is, the paper is too dense, and still too many statements are left undefined or unjustified. (It would make a great journal paper if all of these connections were filled in; I go over specific cases below.) That being said, I think this work would make an excellent conference submission if it were carefully pruned by focusing on only the core development needed to reach the experiments section. E.g. by: removing some of the excess tree description, simply defining the generative process for Mondrian trees (the authors might note the Mondrian process as inspiration without dwelling on the theoretical relations to Mondrian processes---at least not in the main text), defining and using IKN without all of the HPYP and NSP build-up (in particular since HPYP and NSP don't seem to be used in the experiments or for additional insight beyond the IKN connection), etc.

Major points:
* Line 064: Big O in probability is required here, rather than the existing deterministic Big O (cf. https://en.wikipedia.org/wiki/Big_O_in_probability_notation ). If the authors plan to keep this part as is, I recommend they say something brief about what O_p means as it is less standard in this community than Big O.
* Line 078/079: That the distribution of the Mondrian tree is invariant to the order of data points is a statement that needs to be proved (with proof referenced here), at least in the supplementary material.
* Line 081: The authors state the depth of the tree is "typically logarithmic in n." Elsewhere, they state complexity bounds for balanced trees. This depth statement needs to be justified if it remains in the manuscript. Ideally, the authors would demonstrate (with informative summaries and/or graphs and/or tables) how balanced the trees were that were found in the experimental runs. Again, this could happen in the supplementary material but should be referenced when it is used in the main text. If the trees are not uniformly balanced, it is worth providing an analysis for other cases as well, if possible.
* Section 3: A long time is spent reviewing trees in general and rooted, binary trees in particular. This wouldn't be so bad except that short shrift is given to Mondrian processes and the definition of Mondrian trees. Readers in this audience are likely familiar with rooted, binary trees but are less likely to be familiar with Mondrian processes, which are not defined in this manuscript.
* Line 159--163: Somewhere in this area it should be noted that one expects the model to be very sensitive to scaling (due to the role of the "linear dimension" here) and that the authors recommend scaling each dimension to [0,1]. Currently, [0,1]-scaling is just an offhand remark in the experiments section.
* Line 169--174: Bayesian CART (Chipman, George, McCulloch 1998) should be mentioned here as well since variants of these ideas have already been used in that model.
* Line 174--185: If mentioned, projectivity needs a proof, perhaps in the supplementary material. It's not immediately clear that the data dependence in MTs doesn't spoil projectivity but could be established within a few lines.
* Line 203--207: If the authors keep the discussion of NSPs and PYPs, they should provide citations for these models when they are introduced.
* Footnote 2: The stated relative tractability of NSPs needs a citation here.
* Figure 2 caption: Half a page is too much text for a caption, which as a rule of thumb should not generally exceed one paragraph. The vast majority of this text should be in the main text---or at worst the supplementary material, despite the caption's appealing small font size.
* Algorithm 4: It is worth noting somewhere in this area that the reason this extension works, at a high level, is the property of exponential distributions that: X ~ Exp(a), Y ~ Exp(b) implies min(X,Y) ~ Exp(a+b)
* Section 6: Bayesian CART, and possibly related work such as BART, should appear at least briefly in the related work section.
* Line 360: It would be good to see a citation for the stated popularity of this choice of M.
* Line 361: Since \lambda = \infty in the experiments presumably implies that each data point is in its own leaf, this might be a chance to prune some discussion of \lambda from the manuscript and thereby shorten it. Alternatively, if the authors keep the discussion of \lambda, the authors should note the implications in the text of choosing \lambda = \infty.
* Line 361: \gamma = 10 D: It would be helpful to use names for all parameters on this line instead of just symbols. Even better might be to provide a reference to where they were defined in the manuscript. \gamma is somewhat buried on line 210 as is.
* Line 364: Why not run more random initializations and report error bars for each algorithm as well? If the error bars are so small as to not appear on the plot, the authors could mention this in the text or caption.

Minor editing points:
* Line 053: It seems like D_i should be (X_i, Y_i). The existing notation for D_{1:n} is clunky and doesn't degrade well for subsets of 1:n.
* Line 106: "limit" -> "bounds"
* Line 194: "encourages label" -> "encourages the label"
* Line 194: "discuss the" -> "discuss how the"
* Line 238: "rectngble" -> "rectangle"
* Line 360: "As common" -> "As is common"
Summary: The model in this paper is interesting, the experiments are convincing, and the application to streaming data seems timely and important. My major qualm with this manuscript is that it reads as though the authors have tried to fit a journal paper into the space allotted by NIPS.
Author Feedback
Author rebuttal: We thank the reviewers for their detailed and constructive feedback. We will try our best to incorporate everyone’s suggestions into the final version of the manuscript.

Our goal in this paper was to combine the theoretical properties of Mondrian process and the excellent empirical performance of (batch) random forests to create a first-of-its-kind online random forest. Your positive comments are much appreciated. We believe that this work will have a major impact on the online RF literature, and will come to represent an important step towards delivering a powerful predictor for real-world streaming data.

Specific comments:

R_15: The concern about high dimensional data and irrelevant dimensions is valid; however, it was our thinking from the beginning that this problem was beyond the scope of this paper. Understanding the possibilities/limitations of online random forest methods in this challenging setting is a key direction of research that we are hoping this paper inspires other researchers to tackle with us. As we pointed out in the experiments, the accuracy of MF drops if there are too many irrelevant features. Of course, this problem is not unique to MF; ORF-Saffari performs worse than MF on the dna dataset in Fig 3. In batch mode, one could address this problem using the SMC framework proposed in [11] to incorporate the response variable. We suspect there will be many challenges translating these ideas to the online setting. For high-dim data where features are both relevant and uncorrelated, one way to reduce computational cost would be to combine MF with a random subspace method, e.g., Ho, T. K. (1998) “The random subspace method for constructing decision forests”. To the best of our knowledge, the performance of online random forests has not been systematically evaluated on high dimensional data with varying feature relevances. These situations and their solutions may well be highly problem specific; we hope that this paper will inspire more work in this direction.

R_2: We agree that a brief discussion about BMA vs ensemble combination would be helpful. Your feedback is welcome, but we would be inclined to highlight the assumptions that one is required to make for BMA to optimal, and highlight the differences with ensemble combination.

We agree that ruling out dynamic trees was premature. In response to your comment we ran some preliminary experiments evaluating dynamic trees in an “online” setting using the dynaTree R package. (Note that while dynamic trees employ ‘linear leaves’ or ‘constant leaves’ for regression tasks, they use ‘multinomial leaves’ for classification tasks.)

We set n_particles=100, n_passes=R=2 (the code doesn’t seem to allow n_passes=1) and set the remaining parameters to their default values. Here is test accuracy as a function of the fraction of training data (setup similar to Fig 3 is):
usps: [ 0.737 0.776 0.791 0.804 0.814 0.822 0.824 0.833 0.837 0.842]
satimage: [ 0.796 0.805 0.825 0.826 0.837 0.839 0.834 0.84 0.842 0.846]
letter: [ 0.606 0.692 0.725 0.748 0.763 0.784 0.798 0.805 0.817 0.823]
dna: [ 0.768 0.858 0.864 0.879 0.866 0.882 0.898 0.906 0.902 0.904]
We observe that MF achieves significantly higher test accuracies than dynamic trees on usps, satimage and letter. On dna dataset, dynamic trees outperform MF (indicating the usefulness of using labels to guide splits) --- however, MF on only the 60 most “relevant” features (MF^\dagger) achieves similar performance as dynamic trees. Perhaps we are old fashioned, but we think it would be wise to perform a more rigorous experimental study with peer review before publishing any claims about dynamic trees though. Feedback welcome.

R_36: Your extensive comments are very much appreciated. Given space limitations, we can only make a few comments.

Sensitivity to scaling the features: Yes, we assume that all features have been scaled appropriately. While ORF-Saffari makes essentially the same assumption (in order to choose sensible candidate split locations where both children are non-empty), we agree it would be worthwhile to search for methods that do not. This is yet more research to pursue!

Re: balanced trees. We computed the average depth of the trees in the forest (where depth of a leaf node is weighted by fraction of data points at that leaf node):
usps: log2(N)=12.8, depth=19.1+-1.3
dna: log2(N)=10.5, depth=12+-0.3
letter: log2(N)=13.9, depth=23.2+-1.8
satimages: log2(N)=11.6, depth=17.4+-1.6
The values suggest that the depth of the forest scales as log(N) rather than N, but this deserves more attention.

Readability and organization: This paper has undergone several iterations with feedback from colleagues. An earlier version, which did not lead up to Mondrian Forests first by building up notation for decision trees confused several readers who were decision tree experts but not probabilistic modeling experts. Likewise, we considered describing “IKN without all of the HPYP and NSP build-up”, but then the dependence on the edge lengths seems like a hack. Your suggestion about simplifying the presentation based on lambda=infinity is interesting. That said, it is difficult to understand how to handle leaf nodes and the requisite smoothing without knowing the guiding principles. We tried to simplify the text such that it contains principled model definition as well as sufficient amount of intuition. While some of the text could be tailored to the specific experimental setup in this paper, we believe that the probabilistic foundations are useful for others interested in building on this work. We provide detailed pseudocode for those interested in re-implementing the algorithm (our python implementation will be made publicly available as well).